# Defect-mediated ripening of core-shell nanostructures

Qiubo Zhang[1,8], Xinxing Peng [2,8], Yifan Nie[1], Qi Zheng[1], Junyi Shangguan[1,3], Chao Zhu [4], Karen C. Bustillo [2], Peter Ercius [2], Linwang Wang [1], David T. Limmer[1,5,6,7] & Haimei Zheng [1,3✉]

Understanding nanostructure ripening mechanisms is desirable for gaining insight on the growth and potential applications of nanoscale materials. However, the atomic pathways of nanostructure ripening in solution have rarely been observed directly. Here, we report defect-mediated ripening of Cd-CdCl$_2$ core-shell nanoparticles (CSN) revealed by in-situ atomic resolution imaging with liquid cell transmission electron microscopy. We find that ripening is initiated by dissolution of the nanoparticle with an incomplete CdCl$_2$ shell, and that the areas of the Cd core that are exposed to the solution are etched first. The growth of the other nanoparticles is achieved by generating crack defects in the shell, followed by ion diffusion through the cracks. Subsequent healing of crack defects leads to a highly crystalline CSN. The formation and annihilation of crack defects in the CdCl$_2$ shell, accompanied by disordering and crystallization of the shell structure, mediate the ripening of Cd-CdCl$_2$ CSN in the solution.

[1] Materials Science Division, Lawrence Berkeley National Laboratory, Berkeley, CA 94720, USA. [2] National Center for Electron Microscopy, Molecular Foundry, Lawrence Berkeley National Laboratory, Berkeley, CA 94720, USA. [3] Department of Materials Science and Engineering, University of California, Berkeley, CA 94720, USA. [4] School of Materials Science and Engineering, Nanyang Technological University, Singapore 639798, Singapore. [5] Department of Chemistry, University of California, Berkeley, CA 94720, USA. [6] Chemical Science Division, Lawrence Berkeley National Laboratory, Berkeley, CA 94720, USA. [7] Kavli Energy Nanoscience Institute, Berkeley, CA 94720, USA. [8] These authors contributed equally: Qiubo Zhang, Xinxing Peng. ✉email: hmzheng@lbl.gov

Ripening of nanostructures frequently occurs in the growth of nanocrystals[1,2], nanoporous structure formation[3,4], heterogeneous catalysis[5,6], and other nanoscale materials processes[7,8]. Ostwald ripening is most commonly reported, in which bigger nanoparticles grow at the expense of smaller nanoparticles to minimize the total free energy[1,9]. A variety of factors may impact this process, including surfactants[10,11], defects[12], ion concentration[13], and gas molecule adsorption[14]. For example, by using appropriate surfactants to modify the surface energy of nanoparticles, "digestive ripening" can be achieved where smaller nanoparticles grow by consuming bigger nanoparticles, which results in homogenous particle sizes[15]. To date, many ripening processes are only described thermal dynamically; however, the detailed atomic pathways of ripening remain elusive. At the nanoscale, heterogeneity and fluctuations often dominate[16,17]. Defects also play a significant role in many physical and chemical processes of nanoscale materials. Unfortunately, how defects impact the atomic pathways of ripening, especially in solution processes, has rarely been reported. This is largely due to the challenges in tracking nanomaterial dynamics with atomic resolution.

Core-shell nanocrystal system has core and shell materials with distinct crystal structures allowing differentiation of them in TEM images, clear imaging of the morphology dynamically, and tracking of mass transfer. Furthermore, the ripening of core-shell nanoparticles involves the mass transfer between the cores of two nanoparticles. As the core is encapsulated by the shell, an investigation of the core-shell nanoparticle ripening reveals the atomic reconstruction of the whole nanoparticle during ripening.

Here, with cadmium-cadmium chloride (Cd-CdCl$_2$) CSN as a model system, we study the ripening of colloidal nanocrystals with a focus on the role of defects. We track the structural evolution of Cd-CdCl$_2$ core-shell nanoparticles by taking advantage of the recent advances in liquid cell transmission electron microscopy (LC-TEM)[18–21]. With thin carbon films as the liquid cell membrane, atomic resolution imaging is achieved, which provides opportunities to reveal defect-mediated ripening of Cd-CdCl$_2$ core-shell nanocrystals. In this system, ripening proceeds through the formation and annihilation of crack defects, accompanied by mass transfer through the cracks. Resolving the role of defects in nanostructure ripening sheds light on a wide range of nanoscale phenomena.

## Results and Discussion

**Experimental setup**. We conducted the experiments in a home-made carbon film liquid cell (Fig. 1a; also see Methods). The carbon film cell produces small liquid pockets, confining a few nanoparticles in a thin liquid film (10–50 nm in thickness) in individual pockets. The Cd-CdCl$_2$ core-shell nanocrystals are formed in-situ under electron beam irradiation (Supplementary Fig. 1 and Methods)[20]. The electron beam acts as an imaging source and a reducing agent that reduces Cd ions to Cd atoms.

As shown in Fig. 1b, the Cd-CdCl$_2$ core-shell particles were imaged from both the side view and the top view. Figure 1c is the side view of a Cd-CdCl$_2$ core-shell nanoparticle along with Cd [10$\bar{1}$0] direction, whereas Fig. 1e is a top view image of another particle from the Cd [0001] direction. The enlarged images from the selected rectangular areas in Fig. 1c, e confirm the crystallinity and orientations of the interfaces (Fig. 1d, f, g). The Cd core shows a hexagonal structure, and the CdCl$_2$ shell is a trigonal phase (Supplementary Fig. 2, 3, Supplementary Table 1). Moiré Patterns with periodicity "M" = 0.66 nm arise from CdCl$_2$ shell being superimposed onto the crystal lattice of the Cd core (Supplementary Fig. 2 and Methods)[22]. This core-shell geometry

implies two interface orientation relationships between the Cd core and the CdCl$_2$ layered shell (Fig. 1h). The {10$\bar{1}$0} facets of Cd are parallel to the {0003} facets of CdCl$_2$, and the {0001} facets of Cd parallel to the {0003} facets of CdCl$_2$. The scanning transmission electron microscopy energy-dispersive X-ray spectroscopy (STEM-EDX) maps and spectrum confirm the Cd-CdCl$_2$ CSN (Fig. 1i, Supplementary Fig. 4).

**The dynamics of defect-mediated ripening**. Figure 2a, b shows the evolution of two Cd-CdCl$_2$ core-shell nanoparticles viewed along the [0001] zone axis during ripening (see Supplementary Movie 1 for detail). Throughout the ripening, nanoparticle 1 (P1) grows at the expense of nanoparticle 2 (P2) until one larger nanoparticle is achieved. The shape evolution of the two cores is shown in contour maps (Fig. 2c).

To better understand how mass transfers between cores and shells, we track the evolution of each nanoparticle separately (Fig. 2d). Based on the changes of the projected areas and the structural features of the particle, we divide the process into four stages. In stage I (0–16 s), only the metal core of P2 (core 2) is oxidized to Cd$^{2+}$ and dissolves in the solution. The projected area decreases from 201 nm$^2$ to 148 nm$^2$, correspondingly. Although core 2 is larger than the core of P1 (core 1) initially, core 2 reacts preferably in the solution due to an incomplete shell of core 2 (0 s) (Fig. 2b; details will be discussed later in the text). From 16 s to 25 s (stage II), mass transfer from the shell of P2 (shell 2) to the shell of P1 (shell 1), leading to a new CdCl$_2$ layer formed on the surface of shell 1 highlighted by a yellow circle (Fig. 2b: 15.7 s). In stage III (25–40 s), crack defects form in shell 1 while the shell expands contributes the growth of core 1, as pointed by the yellow arrow in the enlarged image (Fig. 2b: 34.7 s). By comparing the area changes of core 1 and core 2, it is clear from their anticorrelation that the dissolution of core 2 promotes the growth of core 1. In stage IV (40–50 s), after P2 is completely dissolved, core 1 and shell 1 continue to grow for several seconds until the shell is repaired to form an almost defect-free crystalline structure (Fig. 2d).

Figure 2c shows that only the sides of P1 that are exposed to the solution grow. Although the two particles were in contact with each other (Fig. 2a, b), the mass does not transmit directly through the shared surface. Instead, the core 2 is first etched into the solution, and the growth of core 1 is mediated through the crack defects. Interestingly, the growth of P1 follows a unique path: the core grows directionally along certain facets, and the perimeter of the shell increases accordingly, but the shell thickness is maintained (Fig. 2b, Supplementary Fig. 5). The deviation from the hexagonal shapes of particle 1 is related to cracks, but the relationship is kinetic, not equilibrium. Starting from the hexagonal equilibrium shape, the cracks initiate a transition to the kinetic shape, where the shape is limited by the slowest-growing facets instead of the lowest-energy facets. Our experimental observation shows that the mass transfer between particles does not depend on the size of the particles but on crack defects, which is different from Ostwald ripening[1], digestive ripening[15], intra-particle ripening[23], and merging of core-shell particles[24] (Supplementary Fig. 6 and Methods).

**The impact of crack defects on ripening process**. To quantify the impact of crack defects in the shell, we track the surface structure evolution of each nanoparticle during the ripening process. During the shrinking of P2, we compare the atomic layers etching on different facets. The three facets with incomplete shells are etched much faster than the other three facets with the undamaged shell (see Supplementary Fig. S7 for the

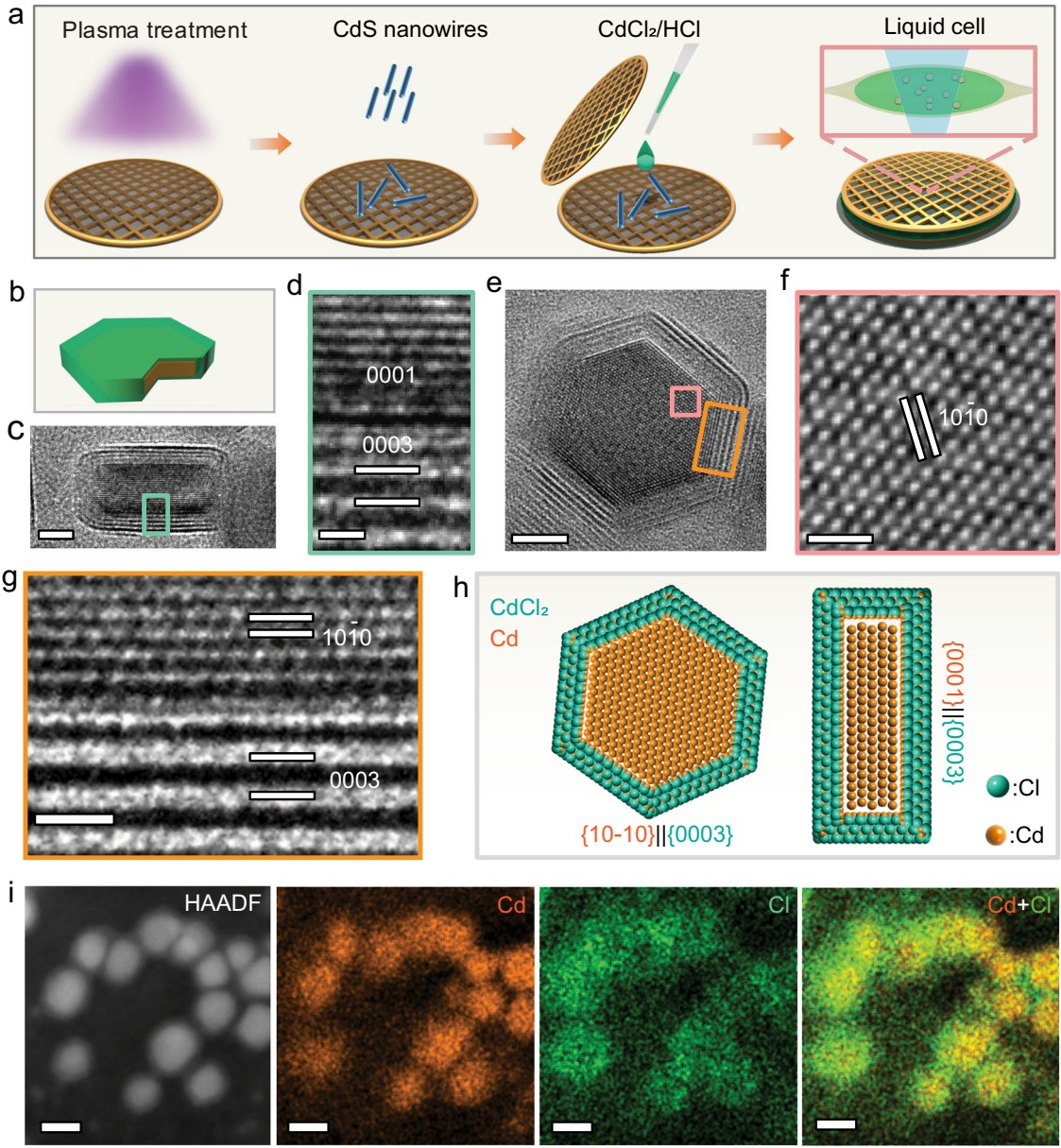

**Fig. 1 Experimental setup and in-situ characterizations of Cd-CdCl$_2$ core-shell nanostructures. a** A Schematic showing the preparation of a liquid cell and the formation of Cd-CdCl$_2$ core-shell nanoparticles. **b** A model of a core-shell nanoparticle indicating viewing directions for (**c**), (**e**). **c** TEM image from the side view. Scale bar, 5 nm. **d** HRTEM image from the selected area in (**c**) shows the interfacial structure. Scale bar, 1 nm. **e** TEM image from the top view. Scale bar, 5 nm. **f, g** HRTEM images from the selected areas in (**e**) showing the structures of the core and interface, respectively. Scale bar, 1 nm. **h** The schematic atom models of the core-shell structure showing the planar view and side view of the nanostructure with the epitaxial relationship, [(10$\bar{1}$0)Cd// (0003)CdCl$_2$] and [(0001)Cd//(0003)CdCl$_2$]. The Cd atoms are orange and Cl atoms are green in the atom models. **i** High-angle annular dark-field imaging (HAADF) and energy-dispersive X-ray spectroscopy (EDS) mapping of Cd-CdCl$_2$ core-shell structures. Scale bar, 10 nm.

quantification method). Therefore, the shell serves as a protective layer preventing the core from being etched. In Fig. 3a, b, we compare the evolution of (1$\bar{1}$00) and (10$\bar{1}$0) facets at the atomic scale, one facet without a shell and the other with a shell (also see Supplementary Movies 2, 3). For the (1$\bar{1}$00) facet without a shell (Fig. 3a), the edge atoms are quickly removed (0 s). The atoms on the left side exposed to the solution dissolve faster than those on the right side with a partial shell (0–15 s). At the same time, the top layers are also dissolved. Eventually, six atomic layers of the (1$\bar{1}$00) facet have been removed within 25 s. Figure 3c shows that the etching of (1$\bar{1}$00) facet is continuous and nearly linear. However, the dissolution at the (10$\bar{1}$0) facet is different (Fig. 3b, d). Atoms are mainly removed along both sides of the facet, while the middle part

remains (0–25 s). Only after a gap is formed between the core and shell (marked by yellow arrows in Fig. 3b), the top atomic layers at the (10$\bar{1}$0) facet are dissolved quickly (25–28 s). Figure 3d shows that the dissolution of the (10$\bar{1}$0) facet first undergoes a plateau and rapidly dissolves after a gap formation between core and shell. The observation implies that the overall chemical potential drives the P2 to dissolve, while crack defects in the protective shell modulate the effective barrier to dissolve.

The other nanoparticle (P1) in the pair grows during the etching of P2. As with the dissolution of P2, the growth of P1 is mediated by the crack defects. For instance, the four facets with the crack defects grow, and the growth of core 1 is initiated at the onset time of shell rupture. As shown in Supplementary Fig. 8, the

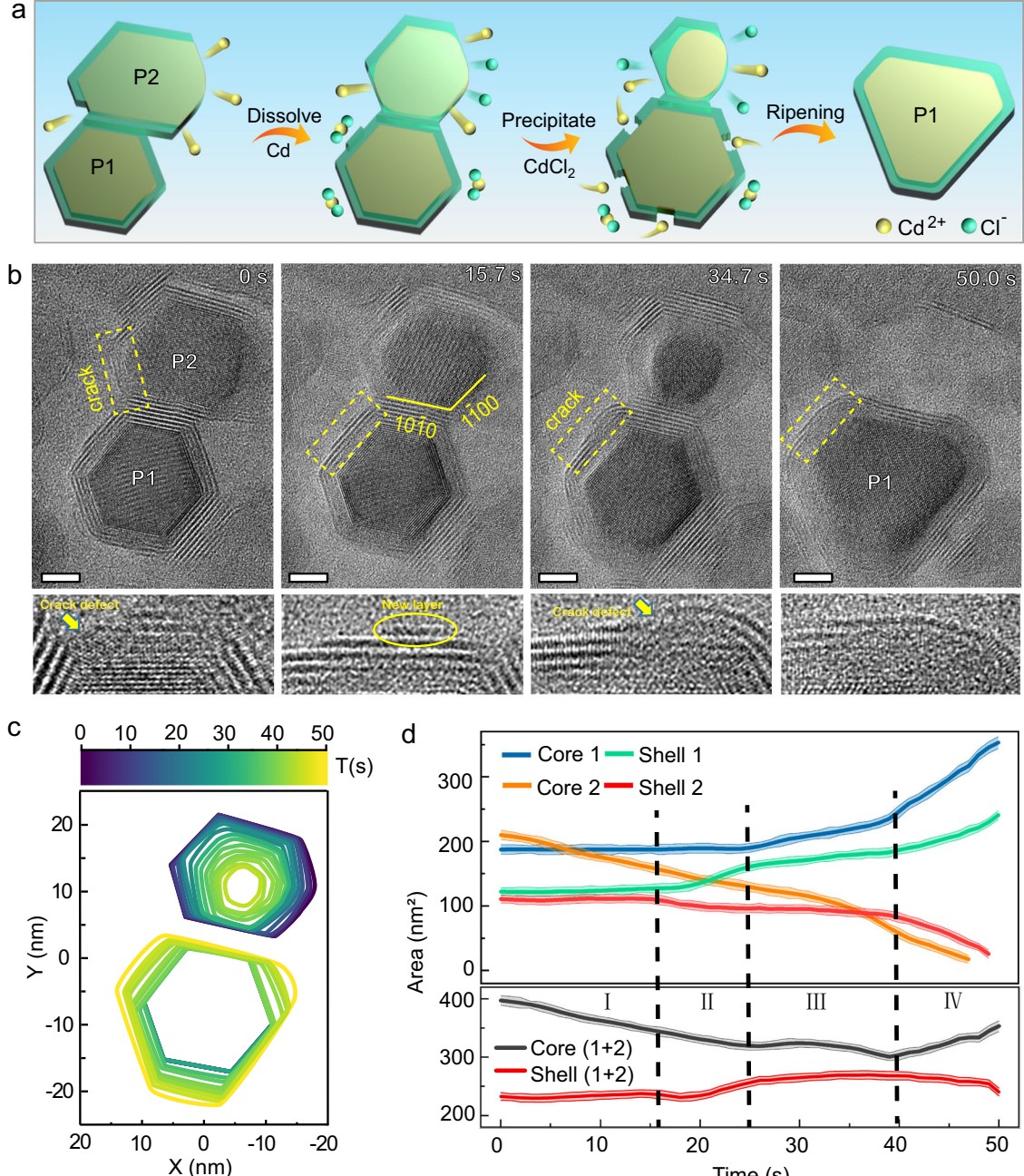

**Fig. 2 The defect-mediated ripening process of Cd-CdCl₂ core-shell nanoparticles. a** Schematic illustration of the evolution pathway of Cd-CdCl₂ core-shell nanostructures during the defects-mediated ripening. **b** Sequential images show the ripening process after two Cd-CdCl₂ core-shell particles are connected. P1 and P2 mark out the two particles. Scale bar, 5 nm. The enlarged images of the yellow square areas listed below highlight an incomplete shell, a new CdCl₂ layer, a new crack defect and healing of the shell. **c** The colored contours indicate the shape evolution of the two cores. **d** The measured projected areas of the core and shell for P1 and P2 as a function of time. I, II, III, and IV refer to the four intervals of the dynamic process. The transparent lines around the solid lines indicate the standard deviation error of the measurement.

($0\bar{1}10$) facet starts to grow first, followed by the ($1\bar{1}00$) and ($\bar{1}100$) facets, and finally the ($10\bar{1}0$) facet. However, the other two facets, ($\bar{1}010$) and ($01\bar{1}0$) facets, with intact shells do not grow. Figure 4 depicts the structural evolution of the shell before crack defects formation and how crack defects affect the growth of the ($10\bar{1}0$) facet of the core (Supplementary Movie 4). Distinctly different growth is shown before and after the crack defects formation (24 s). The core height of ($10\bar{1}0$) facet versus time is plotted in Fig. 4a, which shows that the ($10\bar{1}0$) facet has no obvious growth until the crack defects are formed. Then, it grows rapidly

mediated by the crack defects. Figure 4b highlights the structural features in the two stages. In stage I (0–24 s), dynamic changes in the shell can be identified while there is no appreciable growth in the core. A disordered state layer at the Cd/CdCl₂ interface and edge dislocations in the CdCl₂ shells can be found. In stage II (24–40 s), a crack defect is formed in the top shell structure, which serves as a diffusion channel for Cd ions to pass through the shell. Figure 4c reveals the structural evolution before the crack defect formed in the shell. Here, we focus on the top shell on the ($10\bar{1}0$) facet of the core. First, the two layers

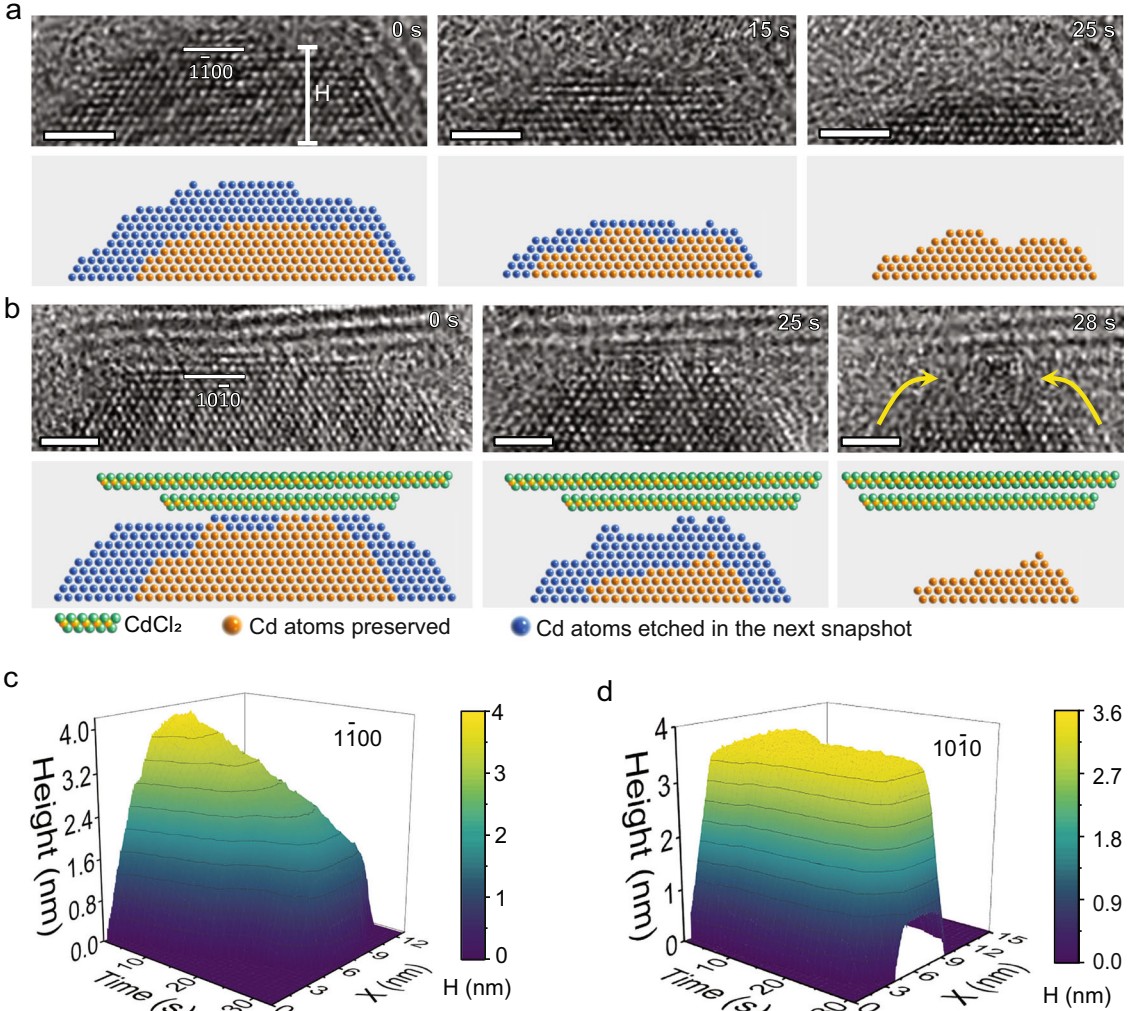

**Fig. 3 Influence of the CdCl₂ shell on the dissolution of the Cd core. a** HRTEM images from Supplementary Movie 2 show the dissolution of Cd (1̄100) facet exposed to the liquid directly. Without the protection of the shell, the atoms can be dissolved in all directions. **b** HRTEM images from Supplementary Movie 3 show the dissolution of Cd (101̄0) facet with a coated shell. Under the protection of the shell, the atoms dissolve mainly from the side. The corresponding atom models are shown below highlight the dissolution of Cd. The green balls are Cl atoms, the orange balls indicate the core Cd atoms preserved and the blue balls indicate the core Cd atoms that are absent in the next snapshot. The yellow arrows point out the gap between core and shell. **c** Height of Cd over time in [1̄100] direction. **d** Height of Cd over time in the [101̄0] direction. In (**c**), (**d**), X represents each position at the bottom of the images. The height is measured as the vertical distance from the Cd surface to the bottom of the images, as marked in (**a**). Scale bar, 2 nm.

(0 and 1st) near the interface change into a disordered state (0 s). Subsequently (15.6–18.5 s), a new CdCl₂ layer (5th) is formed and connected to the segment on the right (5th layer). Interestingly, part of the 3rd layer is disconnected from the 3rd layer and connected to the 2nd layer, forming an edge dislocation (18.5 s). Subsequently, the 4th layer repeats the behavior of the 3rd layer to propagate the edge dislocation upward (20.8 s). At the end, the edge dislocation propagates to the outermost layer and disappears at the 22nd second.

The shell structure fluctuates between crystalline and disordered states, which leads to the generation and annihilation of crack defects (Fig. 4d). We consider that the changes in the shell structure are modulated by fluctuations in the local ion concentration, which is correlated with the formation and healing of crack defects. When the concentration of Cd²⁺ cations in the solution is saturated, driven by chemical potential difference, the Cd²⁺ cations tend to pass through the shell and precipitate on the surface of the Cd core to minimize the total energy of the system. We found ions in the shell are highly mobile. The flux of Cd²⁺ ions passing through the shell promotes the formation of crack

defects. In turn, the generation of crack defects further assists Cd²⁺ ions passing through the shell, reducing the Cd²⁺ ions in the solution. In the end, an almost perfect core-shell structure is achieved (35.4 s). Given the crystallographic relationships between the hexagonal Cd core and the trigonal CdCl₂ shell, there are high-angle grain boundaries at all facet junctions (60° and 90°). Intuitively, one may consider that these grain boundaries are much more likely to form defects than the centers of the facets. However, our experiments show that the defects position is random, crack defects can be healed at one place and regenerated at other place (Fig. 4D). For instance, defects can be found at the grain boundaries (Fig. 4C; 0 s), but these defects are quickly repaired. The mechanisms of crack defects mediated ripening of Cd-CdCl₂ core-shell nanocrystals are discussed further in the following.

In the initial stage, the etching of Cd nanoparticles is driven by oxidation reaction. Due to defects in the shell, the larger nanoparticles (P2) shrink, and the oxidation etching of the Cd core occurs in the area exposed to the solution. The preferential dissolution of Cd is kinetically facilitated by the broken shell. In

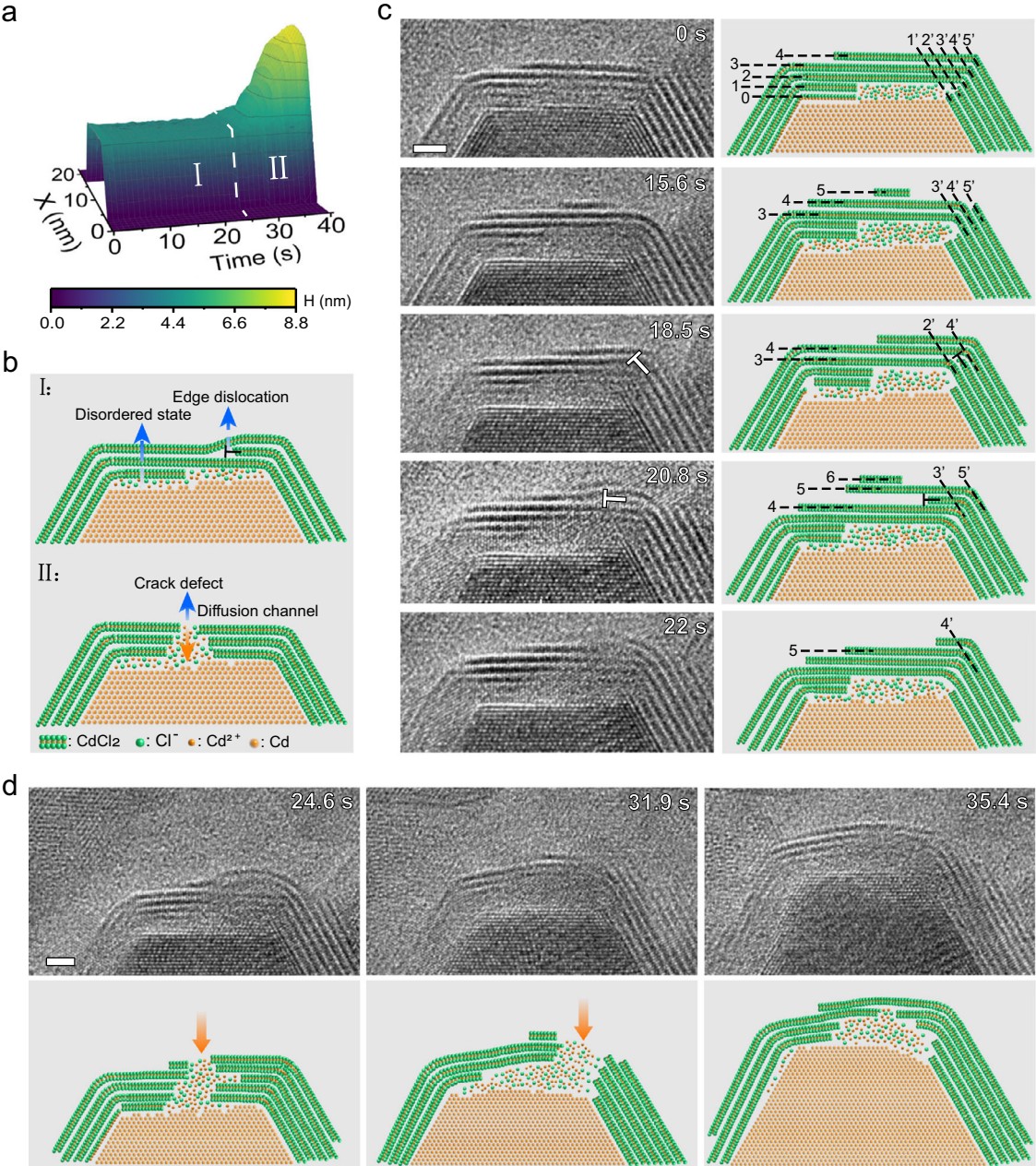

**Fig. 4 The formation of a crack defect and its influence on the directional growth of the Cd core. a** Height of Cd with time in [1000] direction for particle 1. Two distinct stages can be identified: structural evolution before crack defect formation (I) and the crack defect-guided growth (II). **b** The schematic atom models show the structural features of the two stages. In stage I, there is a disordered state layer located at the Cd-CdCl$_2$ interface and an edge dislocation among the CdCl$_2$ shell. In stage II, a crack defect is formed in the top layered structure, which works as a diffusion channel for Cd ions through the shell. Symbol 'T' represents edge dislocation. **c** Image sequences show the generation and propagation of edge dislocation before the formation of a crack defect. Arabic numerals mark the order of the CdCl$_2$ shells. **d** Image sequences show the directional growth of core after shell rupture. The corresponding atom models highlight the details in the formation, propagation of defects, and directional growth of Cd core. The orange arrows point out the crack defects. Scale bar, 2 nm.

the solution, the dissolution of CdCl$_2$ and the etching of Cd are competitive. Because Cd is insoluble in water, if Cd is dissolved in the form of cadmium atoms, the dissolution of CdCl$_2$ will prevail, which is inconsistent with our experimental observations. In our experiment, for we added HCl in the solution, the metal Cd reacted with HCl to generate Cd ions. This is similar to a metal etching process assisted by chloride ions. According to the chemical potential, this chemical process is more advantageous than the dissolution of CdCl$_2$. Thus metal Cd will be oxidized preferentially over CdCl$_2$. The oxidation reactions of Cd can be

described as below, in which Cl$^-$ ions in the solution accelerate the oxidation of Cd to Cd$^{2+}$[25].

$$Cd + 2H^+ + 2Cl^- \rightarrow Cd^{2+} + H_2 + 2Cl^- \qquad (1)$$

Besides H$^+$, other possible oxidants include OH$^•$ or H$_3$O$^+$ from the electrolysis of water. Under electron beam irradiation, radical species, such as e$_h^-$, H$^•$, OH$^•$, H$_2$, H$_2$O$_2$, H$_3$O$^+$, HO$_2^•$, can be generated in the aqueous solution[26,27]. Indeed, the electron beam can generate some oxidants to assist the dissolve of Cd, but Cd can still be dissolved without the electron beam

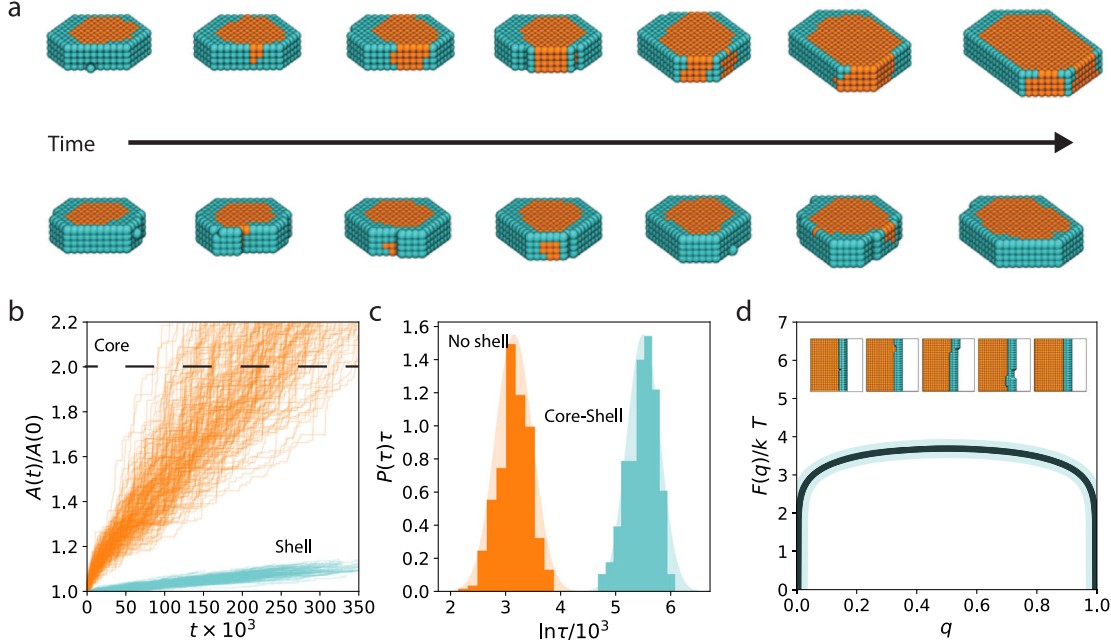

**Fig. 5 Lattice model for defect-mediated growth of core-shell nanostructures. a** Example time series depicting the growth of an initial defect-free core-shell structure with snapshots taken approximately 10000 MC steps apart. **b** Example time series of the projected area of the core and shell over the course of coarsening. **c** Distribution of times for the nanoparticle core to double in size when initialized with and without a shell. **d** Free energy to grow a new layer of core material, proceeding through the reversible generation of a surface crack. The shadings around the solid lines indicate the standard deviation.

(Supplementary Fig. 9). The etching of the Cd metal core increases the concentration of $Cd^{2+}$ ions and further changes chemical potential, which promotes the precipitation of $Cd^{2+}$ and $Cl^-$ after saturation. A new layer of $CdCl_2$ formed on the shell surface of P1 confirmed the saturation of the solution. Once the solution is saturated, the Cd ions in the solution will grow on the surface of core 1 through the formation of crack defects driven by the chemical potential difference.

In order to evaluate the generality of the crack-mediated ripening mechanism, we have developed a simple lattice model. This model can reproduce the basic kinetic processes observed in the liquid cell experiments with the bare physical and chemical components of strong, local interactions and core-shell structure. Figure 5a shows representative trajectories taken from these simulations. The initial nanoparticle persists at a constant size until a fluctuation in the shell exposes the core, after which the core can grow rapidly. If the crack persists, defined here as a fluctuation that exposes the core to the solution, the core will grow until the solution is exhausted of core components. If the shell heals by covering the exposed core, growth stops. Figure 5b shows an example time-series of the area of core and shell, denoted A(t), in which the core grows with the shell nearly unchanged. Figure 5c shows that analogous with the experimental observations, particles without a shell, grow much faster as quantified by the distribution of doubling times P(τ). We note that growth without a shell is still activated, as below the roughening temperature growth proceeds through 2-dimensional nucleation on the surface of the nanoparticle. For the parameters studied, the shell-less particles double in size 20x faster than those with a shell. The importance of the formation of defects on the shell is confirmed in Fig. 5d by computing the free energy F(q) to grow a new layer of the nanoparticle, with order parameter q denoting the progress of formation of a new layer. The free energy has been estimated from brute force simulation by accumulating a histogram of the growth process under conditions where growth and dissolution were equally likely.

A new layer of nanocrystal proceeds reversibly through the spontaneous generation of a defect in the shell at a large energetic cost, here roughly $3k_B T$. The characteristic time to generate this shell defect completely accounts for the separation of growth timescales between the particle with and without a shell, as this time scales as the exponential of the free energy barrier. The presence of a defect that seeds the formation of crack can bypass the requirement of a rare thermal fluctuation to create it spontaneously, and thus can act to catalyze the ripening of the nanoparticle. The $Cd$-$CdCl_2$ system studied experimentally likely has a much higher free energy barrier to the formation of a crack due to the covalent bond energies of the two lattices, admitting a potentially larger catalytic effect of a surface defect. That the basic kinetic processes observed within the LC-TEM have been recapitulated in a highly simplified model suggests that this mechanism is general, and likely to occur in other core-shell nanoparticles.

Understanding the pathways of ripening is important for the controllable synthesis and applications of nanoparticles. As shown in Supplementary Fig. 10, based on the regular hexagonally-shaped $Cd$-$CdCl_2$ core-shell nanoparticles, we can achieve directional growth of one side or multiple sides of particles through crack defects, thereby changing the particle shape. We also propose several promising strategies for controlling the shapes of core-shell structured particles by forming and healing crack defects (Supplementary Fig. 11).

With high-resolution LC-TEM, our direct observation reveals the defect-mediated ripening of $Cd$-$CdCl_2$ core-shell nanoparticles at the atomic scale. At the initial stage of ripening, the nanoparticle with an incomplete shell shrinks. The areas that expose the Cd core to the solution be etched faster. The other nanoparticle starts to grow by generating crack defects and transporting Cd ions through the cracks. The crack defects are initiated by the shell structural fluctuation, which accompanies the formation and propagation of edge dislocations. In the end, healing of the crack defects results in a highly crystalline

nanoparticle. This study deepens our knowledge of ripening in nanostructures and sheds light on the governing factors in nanoscale dynamic processes.

## Methods

**Chemicals**. All commercially available chemicals including cadmium acetate dehydrate (Cd(CH$_3$COO)·2H$_2$O; 99.99%, Aladdin), Sublimed sulfur (S; 99.95%, Aladdin), ethylenediamine (NH$_2$CH$_2$CH$_2$NH$_2$; 98%, Aladdin), methanol (CH$_3$OH; 99.9%, Sigma-Aldrich), ethanol (CH$_3$CH$_2$OH; 99.5%, Sigma-Aldrich), cadmium chloride (CdCl$_2$; 99.99%, Sigma-Aldrich), and Hydrochloric acid (HCl; 37%, Sigma-Aldrich). The growth solution was prepared by dissolving CdCl$_2$ (5 mM) and HCl (1 mM) in deionized water.

**CdS nanorods preparation**. CdS nanorods were synthesized via a hydrothermal method[28]. In brief, 0.2665 g of Cd(CH$_3$COO)·2H$_2$O and 0.0641 g of sublimed sulfur were dissolved in 40 ml ethylenediamine under vigorous stirring and then transferred to a Teflon-lined stainless steel autoclave. The autoclave was sealed and heated at 200 °C for 2 h after which it was allowed to cool to room temperature naturally. The yellowish products were isolated by centrifugation at 5647 g. for 10 min and washed five times with methanol and deionized water.

**In situ liquid cell experimental setup**. We conducted the experiments in a homemade carbon film liquid cell (Fig. 1a). First, the carbon film on a copper grid support was treated with O$_2$ plasma for 10 seconds. Second, a droplet of CdS nanorod solution (0.1 mg/ml) was drop-casted onto a carbon grid and dried off in the air for 5 min. Then, 50 nl of CdCl$_2$ growth solution (5 mM CdCl$_2$ and 1 mM HCl) was dropped onto the CdS-loaded carbon film. We covered the wet grid with another grid. After the liquid cell was assembled, it was loaded into the microscope for imaging. An FEI ThemIS 60-300 STEM/TEM with Cs-corrector was used for in situ imaging, and a beam current density of $1 \times 10^5 - 1 \times 10^6$ e nm$^{-2}$ s$^{-1}$ was maintained for the study.

**Formation of Cd-CdCl$_2$ core-shell nanostructures**. The Cd-CdCl$_2$ core-shell particles are formed in situ by irradiating the growth solution with electron beam[20]. The mechanisms of crystalline nanoparticles formation in liquid cells have been reported in many previous publications[29–31]. Calculations on electron beam induced minimum heating effects have also been reported previously[29–31]. In this work, it is worth mentioning that the formation of the core-shell structures involves the supersaturated precipitation of CdCl$_2$. Part of the cadmium ions are reduced to form metallic cadmium, and some of the cadmium ions are precipitated in the form of cadmium chloride and adsorb on the surface of the cadmium metal to form a core-shell structure. The initial formation process of the Cd-CdCl$_2$ core-shell particles is shown in Supplementary Movie 5 and Supplementary Fig. 1. After they grow up, the shapes tend to be hexagonal nanoplates (Supplementary Fig. 7, 8 and 10). The whole reaction equations are as follows:

$$H_2O \xrightarrow{radiolysis} H^+ + e^- + OH^\bullet \qquad (2)$$

$$Cd^{2+} + 2e^- \rightarrow Cd \qquad (3)$$

**The determination of a CdCl$_2$ shell covering on the (0001) surface of the core**. To illustrate that the basal surface of the particle is covered with a CdCl$_2$ layer, we analyzed the FFT and Moiré Patterns of the particles. The FFT of the yellow box area in the Supplementary Fig. 3 shows two distinct structures: the hexagonal structure Cd and triclinic structure CdCl$_2$, where the <11$\bar{2}$0> direction of the Cd is parallel to the <11$\bar{2}$0> direction of the CdCl$_2$. The presence of Moiré Patterns further confirms the existence of the CdCl$_2$ shell. Supplementary Fig. 3 shows an obvious Moiré Patterns with periodicity "M" = 0.66 nm, the corresponding diffraction points are marked by a red line. According to the principle of the Moiré Patterns formation[32], we can calculate the Moiré Patterns formed by (11$\bar{2}$0) facets of Cd and (11$\bar{2}$0) facet of CdCl$_2$. As shown in Supplementary Fig. 2, the interspace of Cd (110) facets is $D_1 = 0.14885$ nm, the interspace of CdCl$_2$ (11$\bar{2}$0) facets is $D_2 = 0.192295$ nm. The spacing of the Moiré Patterns is $M_c$ given by[33]:

$$M_c = \frac{D_1^2}{D_2 - D_1} + D_1, \qquad (4)$$

The calculated value ($M_c = 0.659$ nm) is in good agreement with the observation ($M = 0.66$ nm). This agreement in the measured Moiré Patterns suggests that the CdCl$_2$ shell lies on the (0001) facet of the core.

**Metropolis Monte Carlo simulation**. We have developed and studied a minimal lattice model to understand the generality of crack-mediated ripening mechanism. This model is capable of reproducing these basic kinetic process observed with only the bare physical and chemical components of strong, local interactions and core-shell structure. Specifically, we considered a two-component lattice model defined on a three-dimensional lattice with local hexagonal symmetry. The instantaneous state of the model can be represented by a string of N occupation variables for each

*i*'th lattice site, such that $n_i = \{0, 1, 2\}$, where the occupation variable $n_i$ denotes either an empty state $n_i=0$, a core state $n_i=1$ or a shell state $n_i=2$. The energy function for this system is of the form of a Potts model[34],

$$E = -\sum_{<i,j>} \epsilon_{n_i,n_j} n_i n_j - \sum_{i=1} \mu_{n_i} n_i \qquad (5)$$

where the first sum is restricted to nearest neighbors, and

$$\epsilon_{n,m} = \begin{cases} \epsilon_c & n = m = 1 \\ \epsilon_s & n = m = 2 \\ \epsilon_i & n \neq m > 0 \end{cases} \qquad (6)$$

are the interaction energies for the core-core, shell-shell, or core-shell respectively, while

$$\mu_n = \begin{cases} \epsilon_c & n = 1 \\ \epsilon_s & n = 2 \end{cases} \qquad (7)$$

are the chemical potentials for the core and shell, respectively. We find that choosing $\epsilon_c > \epsilon_s = \epsilon_i$ with $\epsilon_c/k_BT$ above the roughening coupling strength[35] relative to Boltzman's constant times temperature $k_BT$ is capable of stablizing a core-shell structure when the number of initial shell sites are much smaller than the number of core sites and both are held constant in a canonical ensemble. For our simulations we employ $\epsilon_c/k_BT = 10$, $\epsilon_s/k_BT = \epsilon_i/k_BT = 6.7$, reasonably consistent with the adsorption energy scales computed from the DFT calculations (Supplementary Fig. 12).

We have simulated this model using Metropolis Monte Carlo dynamics in a grand canonical ensemble[36] in which both the total number of core and shell sites are allowed to fluctuate in response to chemical potentials $\mu_c/k_BT = -30$ and $\mu_s/k_BT = -20$. The use of a grand canonical ensemble admits a bath of potential core and shell sites which an initial nanocrystal can take up to coarsen and models the effect of a dissolving nanoparticles in its vicinity. The large values reflect the expected sparing solubility of the core and shell components. A lattice of $N=80 \times 80 \times 4$ sites is used for all simulations with periodic boundary conditions, and an initial hexagonal nanocrystal containing 200 core sites and a 2 layer thick shell site acts as a conserved initial condition as illustrated in Fig. 5a. The Monte Carlo dynamics include nearest eight swap moves and insertion/deletion moves at a ratio of 1:50 in order to model the facile equilibration of the solution. A time-step is considered $N$ attempted Monte Carlo moves.

**Estimation of ion concentration in the solution**. The estimation of ion concentration in the solution is based on changes in the shape of the two particles. First, we directly observed changes in the projected area of the two particles during the whole process (Fig. 2d). In Supplementary Fig. 12, we measured the thickness of the core and the whole particle. We assumed that the thicknesses of the two cores and the two particles are equal. The core is 7 nm thick and the nanoplate is 12 nm thick. During the dissolution process, the thickness of the nanosheets hardly changed in the previous stage (Supplementary Fig. 13). Therefore, the projected area can reflect the change in the volume. According to the volume equation: $V = S \times T$, where $V$ is volume, $S$ is projected area, and $T$ is thickness. We can calculate the volume of each part of the two particles, including core 1, core 2, shell 1, and shell 2. Next, we estimated the thickness, measured the area of the liquid pocket, and calculated the volume of the solution. Our estimated volumes of the liquid pack was 50,000 nm$^3$. We regarded the 16$^{th}$ second as the saturation point of the solution when the cadmium chloride had just begun to precipitate. Then we calculated the ions concentration in the solution at each moment based on the change of volume of the two particles (Supplementary Fig. 14). The density and molar mass of Cd and CdCl$_2$, saturation ion concentration of Cl$^-$ and Cd$^{2+}$, and all parameters related to the calculation are shown in Supplementary Table 2.

**The comparison of defect-mediated ripening with other ripening processes**. The ripening phenomenon previously reported is a thermodynamically driven process in which mass is transferred from high-energy surfaces to low-energy surfaces. For example, in Ostwald ripening[1], the surface energy of small particles is larger. Mass transfer from small particles to large particles, reducing the total energy of the system. In digestive ripening[15], due to the modification of ligands, the larger particles have larger surface energy. Mass transfer from larger particles to smaller particles results in homogenous particle sizes. For core-shell structured particles, it is hard to imagine how two or more core-shell structured particles become one core-shell structure without generating and repairing the crack defects. If only considering thermodynamics, the shell of small particles might first dissolve in the solution and precipitate on the surface of large particles, making the shell of large particles thicker. Then the core of large particles begins to dissolve, nucleate and grow on the surface of the large particle, finally forming a core-shell-shell structure. But our observation is different. The existence of defects in the shell makes the solid-liquid interface between the solution and the particle different, resulting in a difference in dissolution kinetics. Therefore, in our observation, the mass transfer is not completely dependent on the size but is dominated by the crack defects in the shell. When the solution is supersaturated, driven by thermodynamics, Cd metal will not nucleate on the surface of the CdCl$_2$ shell. Instead, it prefers to break the shell to promote the growth of the core. Combined with the structural fluctuation of the shell, crack defects are generated under the impact

force of foreign ions. Then foreign metal ions can enter through crack defects to promote the growth of the core. At the end of ripening, the crack defects can self-heal, lead to a complete core-shell structure. Without direct observation, one might hard to imagine the unique ripening pathway of core-shell particles through the generation and healing of crack defects.

**Defect-driven growth can achieve unique growth**. Based on defect-mediated ripening, we propose a defect-mediated growth mode of core-shell structured particles (CSP), which can be used to guide the controllable synthesis of CSP (Supplementary Fig. 10 and 11). As shown in Supplementary Fig. 5, the two approaches for classic growth of CSP are bottom-up[37] and top-down[38]. In the bottom-up growth methods, a core is usually used as a precursor material, and a shell is formed on the surface of the core and then grows radially outward. During the growth, the core size remains, and the shell continues to thicken. For the top-down method, a shell is formed beneath the core surface by oxidation or ion exchange reaction, and then it grows inward. In this case, the outward layer increases meanwhile consuming the core, resulting in a thicker shell and a smaller core. However, whether it is a bottom-up or top-down approach, directly growing the core of CSP is challenging because the outer layer prevents the penetration of the growth material. Interestingly, in defect-mediated growth, the growth of the CSP follows a different pathway: the core keeps growing through crack defects, the perimeter of the shell increases correspondingly, but the shell thickness is almost unchanged (Supplementary Fig. 5).

**Exclude the influence of surrounding particles on the ripening of the nano-plate pair**. We show that the surrounding particles have little influence on the ripening process of the nanoparticle pair in our study from the following points. First, unlike silicon nitride cells that form large liquid pools, carbon film cells form many small liquid pockets that enclose a small number of particles. As shown in Supplementary Fig. 12, the liquid is mainly distributed in the yellow frame region, and only the two particles we focus on are completely within it. Secondly, although some other particles are partially in the liquid, due to the distance between particles and the limitation of diffusion rate, the mass transfer is mainly from particle 2 to its nearest neighbor (particle 1). In the movie, we observed that the shapes of other particles did not change significantly, and the mass transfer was mainly carried out between particle 1 and particle 2. Moreover, as shown in Fig. 2c, after the solution is saturated, the total projected area of the two particles is basically the same before and after the ripening. Furthermore, in previous studies, multiple particles exist in the solution inevitably. It is reasonable and acceptable to focus on two adjacent particles to study the ripening phenomenon. Therefore, the influence of other surrounding particles on defect-mediated ripening is considered not significant.

**The influence of electron beam on the formation of cracks**. The electron beam indeed affects the etching of Cd by improving the etching speed, but it is not the main driving force for the formation of defects in our observation. We have col-lected a video (Supplementary Movie 6) before the ripening of particles. Under the same beam conditions, P1 does not produce cracking defects under the electron beam irradiation for an extended period of time before P2 starts to dissolve. This suggests that electron beam irradiation is not the driving force for defect formation, but the chemical potential changes resulting from the particle dissolution. From the Monte Carlo simulation, we can see even if the electron beam effect is not taken into account, the crack defects still spontaneously occur. Therefore, we conclude that e-beam is not the main factor for crack generation in this experiment.

## Data availability

All TEM data supporting the findings of this study are contained in the paper and its Supplementary Information files. All other relevant data are available from the corresponding author (H. Zheng) on request.

## Code availability

All codes supporting the findings of this study are available from the corresponding authors upon request.

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

## Acknowledgements

This work was supported by the U.S. Department of Energy, Office of Science, Office of Basic Energy Sciences (BES), Materials Sciences and Engineering Division under Contract No. DE-AC02-05-CH11231 within the in-situ TEM program (KC22ZH). Work at the Molecular Foundry of Lawrence Berkeley National Laboratory (LBNL) was supported by the U.S. Department of Energy under Contract No. DE-AC02-05CH11231.

## Author contributions

Q.Z. conceived and H.Z. supervised this project. Q.Z. performed the synthesis. Q.Z. and X.P. designed and performed the in situ TEM imaging. Y.N. performed the density functional theory calculation under the supervision of L.W. D.T.L. did the metropolis Monte Carlo simulation. Q.Z., C.Z., and Qi. Z. drew the schematic diagrams. Q.Z., X.P., K.B., J.S., P.E., L.W., D.T.L., and H.Z., carried out the data analysis. Q.Z. and H.Z. co-wrote the manuscript with input from all authors.

## Competing interests

The authors declare no competing interests.
