## [Peer Review File · Nature Communications]

Defect-mediated ripening of core-shell nanostructuresREVIEWER COMMENTS

Reviewer #1 (Remarks to the Author):

This manuscript describes a detailed investigation of the growth process of Cd/CdCl₂ nanoparticles. The microscopy work presented here is very impressive and provides a very detailed picture of the underlying processes. In particular, the observation of the importance of defects in the CdCl₂ shell is very interesting, although it is unclear to which the very specific details of this system are transferable to other systems.

Unfortunately, the discussion section does not meet the otherwise high quality of this work. Apart from being in parts difficult to read, see, e.g., the sentence on lines 172-174 (what is a "heterogeneity difference"?), the arguments presented are poorly and/or incompletely backed up. In particular the analysis of the density functional theory calculations is not suitable to support the discussion. The authors have calculated adsorption energies (see Eq. (5)) using atomic reference states, neglecting entirely the ionic character of the ion source (in solution), solvation as well as the role of chemical potentials/concentrations, even though they repeatedly refer to these quantities in the discussion. There is an extensive literature on the so-called ab-initio thermodynamics of adsorption processes on surfaces in contact with both gases and liquids, which would be a valuable resource here.

I am also confused by the discussion of Cd vs CdCl₂ in solution. I am not familiar with Cd solvation specifically, but at least in the case of Au, which has been extensively studied, it is usually not Au⁺ or Au³⁺ that is present in solution but rather AuCl₂⁻ or AuBr₂⁻ or similar. These aspects are rather crucial for quantitatively understanding the role of chemical potentials and their coupling with the ion concentrations in the solution.

Related to the last point, the analysis of the Cd²⁺ and Cl⁻ concentrations (SI Figure 9) and Appendix is interesting and very relevant. I did not understand why two different volumes were considered.

Overall I consider this to be a very interesting study of an important subject that could be of interest for publication in Nature Communications. In the current version there are, however, severe shortcomings that need to be addressed.

Reviewer #2 (Remarks to the Author):

In this work, the authors report defect-mediated ripening of Cd-CdCl₂ core-shell nanoparticles (CSN) using in-situ atomic resolution imaging with liquid cell transmission electron microscopy. The ripening was achieved by generating crack defects in the shell, followed by ion diffusion through the cracks. The healing of crack defects led to a highly crystalline CSN. The formation and annihilation of crack defects in the shell, accompanied by disordering and crystallization of the shell structure, mediate the ripening of CSN in the solution. This process is a kinetic controlled process. The defect-mediated ripening pathway is a new growth pathway and was rarely reported previously, but it is critical for understanding crystal growth. This manuscript is well organized, clear, and concise. Therefore, I recommend acceptance after minor revision.

Comments:

For a deeper understanding, statistical analysis of the formation of the cracks is needed. For example, when the cracks form, where do they prefer to form, at the corner, middle of the facets, or it's random? Does the site affect the ripening and growth? How does the e-beam dose rate affect the formation of cracks?

Reviewer #3 (Remarks to the Author):

The authors image liquid phase digestive ripening process in Cd/CdCl₂ core/shell nanoparticles. The results show a novel process, where crack formation in the shell lead to dissolution of nanoparticles and growth of neighboring nanoparticles through a ripening process. The results demonstrate a potentially new mechanism of nanoparticle ripening. My main concerns are that the authors overreach on their understanding of the mechanism and many of their conclusions are not supported by the data, which consists of in situ TEM and DFT data. These are kinetic processes mediated by solution chemistry, which neither in situ TEM or DFT give insight into. I suggest the authors significantly soften their claims in many parts of the manuscript as there is not sufficient evidence for many things they say. These issues should be addressed first in a major revision before considering publication in Nature communications.

1.) The abstract is written in a too general language. The last sentence makes it seem like they have discovered a general mechanism that applies to ripening of all core-shell nanoparticles. Instead, they show results for a very specific type of nanoparticle with specific chemistry in a specific solvent. Its hard to generalize these results to other types of nanoparticles. They should modify the language so it doesn't sound so general to other types of nanoparticles.

2.) In describing Figure 2, the authors ascribe cause to a number of events without any supporting evidence. For instance, the authors state that the core of one of the particles dissolves until the solution is supersaturated. However, the authors don't know whether the halting of particle dissolution is caused by the solution becoming saturated. Similarly, they state that crack initiation leads to enhanced ion diffusion. It is not know that crack initiation is causes enhanced diffusion. I don't dispute that these dynamics are occurring, but the authors should adjust their language so they don't make such strong claims about processes they don't have supporting evidence for. Instead I think a better approach is to discuss these processes together with supporting evidence (other characterization techniques, prior literature, etc). In situ TEM observations alone are not sufficient evidence to make claims about the solution chemistry because it cannot be directly seen.

3.) What causes the dissolution? Clearly the electron beam is causing this, but the authors should address the possible chemical mechanism inducing the crack formation and core dissolution. There are many times in the manuscript where it is implied that digestive ripening is driven by thermodynamics, but I expect that electron beam induced oxidation is driving the dissolution.

4.) In the beginning of the discussion the authors state they estimate the concentration of Cd²⁺ in the solution. How is this done? The statement is kind of non-connected to this section of the discussion, so its unclear why this is mentioned. After this, they state that the low ion concentration in solution promotes Cd dissolution. This simply is not true; Cd is not soluble in the solvent and is thus not subject to thermodynamic driven Cd dissolution. As mentioned previously, I expect that the electron beam is promoting the Cd dissolution through an oxidation process.

5.) In general, the authors make several claims in the discussion about the relative concentration of Cd²⁺ in solution. But as mentioned above they don't know the Cd²⁺ concentration and can only infer it based on the dissolution kinetics of the Cd metal. Further, its more complex than simply looking at Cd metal dissolution kinetics, because one must consider changes in the CdCl₂ layer as well. I think the discussion should be reworded in several places to remove mentions of the relative Cd²⁺ concentration because it cannot be discerned from their microscope images.

Point-to-Point Response to the reviewers' comments

Reviewer #1 (Remarks to the Author):

This manuscript describes a detailed investigation of the growth process of Cd/CdCl₂ nanoparticles. The microscopy work presented here is very impressive and provides a very detailed picture of the underlying processes. In particular, the observation of the importance of defects in the CdCl₂ shell is very interesting, although it is unclear to which the very specific details of this system are transferable to other systems.

We thank this referee for her/his positive comments of our work. We have done significant additional theoretical calculations with a very general lattice model that includes only the basic physics and chemistry of the core-shell structure and strong local interaction. Based on its ability to reproduce the basic qualitative features observed in the Cd-CdCl₂ system, we conclude that our observation of the importance of defects in repining of the core-shell nanoparticle is not specific to the Cd/CdCl₂ system and it should be transferable to other systems.

Comment 1: Unfortunately, the discussion section does not meet the otherwise high quality of this work. Apart from being in parts difficult to read, see, e.g., the sentence on lines 172-174 (what is a "heterogeneity difference"?), the arguments presented are poorly and/or incompletely backed up. In particular the analysis of the density functional theory calculations is not suitable to support the discussion. The authors have calculated adsorption energies (see Eq. (5)) using atomic reference states, neglecting entirely the ionic character of the ion source (in solution), solvation as well as the role of chemical potentials/concentrations, even though they repeatedly refer to these quantities in the discussion. There is an extensive literature on the so-called ab-initio thermodynamics of adsorption processes on surfaces in contact with both gases and liquids, which would be a valuable resource here.

Authors' Reply:

We thank this reviewer for the constructive comments and suggestions. We have made

significant revision of the discussion section and included additional theoretical calculation to support the discussion.

In order to confirm the basic kinetic steps of the ripening mechanism proposed and its generality, we developed a lattice model of a core-shell nanoparticle and employed Monte Carlo simulation. As shown in the new Figure 5 in the revised manuscript, the lattice model containing only the basic core-shell morphology and a description of the thermodynamic driving forces for ripening is able to recapitulate much of the liquid cell TEM observations. An expanded discussion and new methods section details the model and simulation procedures.

Fig. 5: Simulated defect-mediated growth of core-shell nanostructures. **a**, Example time series depicting the growth of an initial defect-free core-shell structure with snapshots taken approximately 10000 MC steps apart. The top row highlights the accelerated growth with defects in the shell, as the contrast to the defect-free core-shell structure at the bottom row. **b**, Example time series of the projected area of the core and shell over the course of coarsening. **c**, Distribution of times for the nanoparticle core to double in size when initialized with and without a shell. **d**, Free energy to grow a new layer of core material, proceeding through the reversible generation of a surface crack.

Comment 2: I am also confused by the discussion of Cd vs CdCl₂ in solution. I am not familiar with Cd solvation specifically, but at least in the case of Au, which has been extensively studied, it is usually not Au⁺ or Au³⁺ that is present in solution but rather AuCl₂⁻ or AuBr₂⁻ or similar. These aspects are rather crucial for quantitatively understanding the role of chemical potentials and their coupling with the ion concentrations in the solution.

Authors' Reply:

Thanks for the comments. Indeed, Au (I/III) ions can easily form stable coordination compounds, like [AuCl₄]⁻. Taking Au (III) as an example, HAuCl₄ is produced by dissolving gold in aqua regia, which contains high concentration of Cl⁻. [AuCl₄]⁻ is stable in water due to the high stability constants of [AuCl_x]^{3-x} complex species (*Journal of Solution Chemistry* **38**, 725-737 (2009)). However, Cd (II) ions have relatively low stability constants for [CdCl_x]^{2-x} complex species compared with Au (*J. Phys. Chem. B*, **117**, 5241–5248 (2013); *J. Phys. Chem.* **64**, 1764–1766 (1960)). In CdCl₂ water solution, Cd²⁺ is dominated when the concentration of CdCl₂ is lower than 0.01 M. In our experiment, we used 0.005 M CdCl₂ water solution, thus Cd²⁺ is expected to be the main species for Cd (II) in the solution. [CdCl₄]²⁻ is expected to dominate, when CdCl₂ is dissolved into a solution with higher concentration of Cl⁻.

Comment 3: Related to the last point, the analysis of the Cd²⁺ and Cl⁻ concentrations (SI Figure 9) and Appendix is interesting and very relevant. I did not understand why two different volumes were considered.

Authors' Reply:

We estimated that the volume of the solution is about 50000 nm³. In order to show that the changing trend of ion concentration is not dependent on the volume of solution, we arbitrarily used an additional volume to show that the changing trend of the concentration is consistent. In the revised manuscript, we removed the additional volume to avoid confusing readers.

Comment 4: Overall I consider this to be a very interesting study of an important subject that could be of interest for publication in Nature Communications. In the current version there are,

however, severe shortcomings that need to be addressed.

Authors' Reply:

Thank you for your comments. We have made point-by-point responses, and believe the revised manuscript have addressed all the concerns of the reviewer.

Reviewer #2 (Remarks to the Author):

In this work, the authors report defect-mediated ripening of Cd-CdCl₂ core-shell nanoparticles (CSN) using in-situ atomic resolution imaging with liquid cell transmission electron microscopy. The ripening was achieved by generating crack defects in the shell, followed by ion diffusion through the cracks. The healing of crack defects led to a highly crystalline CSN. The formation and annihilation of crack defects in the shell, accompanied by disordering and crystallization of the shell structure, mediate the ripening of CSN in the solution. This process is a kinetic controlled process. The defect-mediated ripening pathway is a new growth pathway and was rarely reported previously, but it is critical for understanding crystal growth. This manuscript is well organized, clear, and concise. Therefore, I recommend acceptance after minor revision.

We appreciate the reviewer for the positive affirmation of our work.

Comment 1: For a deeper understanding, statistical analysis of the formation of the cracks is needed. For example, when the cracks form, where do they prefer to form, at the corner, middle of the facets, or it's random?

Authors' Reply:

Based on the experimental observation, the crack defects start to form after the solution reaches saturation, as indicated by the formation of a new CdCl₂ layer (Fig. 2b; 15.7 s). Given the crystallographic relationships between the hexagonal Cd core and the trigonal CdCl₂ shell, there are high-angle grain boundaries at all facet junctions (60° and 90°). One may think that

it is much more likely to form defects at these grain boundaries than the centers of the facets. However, in our experiments, the defects are formed randomly; crack defects can be healed and regenerated at other places (Fig. 4d). For example, we found defects at the grain boundaries (Fig. 4d; 0s), but these defects were quickly repaired.

We did additional Monte Carlo simulation, the results showed that the presence of a defect that seeds the formation of crack can bypass the requirement of thermal fluctuation to create the crack spontaneously, and thus it can catalyze the ripening of the nanoparticle.

Comment 2: Does the site affect the ripening and growth?

Authors' Reply:

Based on observation, the site can affect the ripening and growth of particles. For example, particle 1 has six sides, only four of them directly contacting the solution grow. Therefore, the sites in contact with liquid will take part in the ripening and growing, but the sites that are in contact with the other nanoparticle without being exposed to the liquid solution have almost no change.

Comment 3: How does the e-beam dose rate affect the formation of cracks?

Authors' Reply:

Thanks for the comment. The electron beam effect is a common issue that needs to be understood and controlled for many liquid electron microscopy experiments. Our control experiments show that electron beam indeed affects the dissolution of Cd by accelerating the dissolving speed, but it is not the main driving force for the defects formation.

For instance, we collected a video (added as Movie R1) before the ripening of particles. Under the same beam conditions (Figure R1), particle 1 did not produce crack defects under the same electron beam irradiation for an extended time (20 seconds) before the precipitation of CdCl₂ on the surface of particle 1. This means that electron beam irradiation is not the driving force for defect formation, but it is the chemical potential changes resulting from the particle dissolution. From the Monte Carlo simulation, we can find that without the electron beam

effects, the crack defects spontaneously occur. Therefore, we conclude that e-beam is not the main factor for crack generation in this experiment.

Figure R1. **a**, TEM image of the core-shell particle pair before 20 seconds of crack formation (-20 s). **b**, TEM image of the core-shell particle pair at the time of the precipitation of CdCl₂ (0 s). The initial time is arbitrary.

Reviewer #3 (Remarks to the Author):

The authors image liquid phase digestive ripening process in Cd/CdCl₂ core/shell nanoparticles. The results show a novel process, where crack formation in the shell lead to dissolution of nanoparticles and growth of neighboring nanoparticles through a ripening process. The results demonstrate a potentially new mechanism of nanoparticle ripening. My main concerns are that the authors overreach on their understanding of the mechanism and many of their conclusions are not supported by the data, which consists of in situ TEM and DFT data. These are kinetic processes mediated by solution chemistry, which neither in situ TEM or DFT give insight into. I suggest the authors significantly soften their claims in many parts of the manuscript as there is not sufficient evidence for many things they say. These issues should be addressed first in a major revision before considering publication in Nature communications.

We thank this reviewer for the comment. We have made major revision of the manuscript and included the additional theoretical calculation to support the discussion of kinetic processes observed in the experiments. In the revised manuscript, we can assure the readers that all the claims are supported by the experiments.

Comment 1: The abstract is written in a too general language. The last sentence makes it seem like they have discovered a general mechanism that applies to ripening of all core-shell nanoparticles. Instead, they show results for a very specific type of nanoparticle with specific chemistry in a specific solvent. Its hard to generalize these results to other types of nanoparticles. They should modify the language so it doesn't sound so general to other types of nanoparticles.

Authors' Reply

To evaluate the generality of the observed crack-mediated ripening mechanism, we developed a lattice model of a core-shell nanoparticle and employed Monte Carlo simulation. As shown in the new Figure 5 in the revised manuscript, the lattice model containing only the basic core-shell morphology and a description of the thermodynamic driving forces for ripening is able to recapitulate much of the liquid cell TEM observations. The fact that the basic kinetic processes observed in the liquid cell TEM experiments have been recapitulated using such a simplified model suggests that this mechanism is general. We consider the crack-mediated ripening mechanism may apply to other core-shell nanoparticles. For example, previous researches (*J. Phys. Chem. C* 2013, 117, 20043–20053; *Nature communications* 2020, 11:3041) have reported that crack defects in the shell can influence the etching of core-shell structures.

We have modified the abstract by specifying the system of Cd-CdCl₂ in this study.

Comment 2: In describing Figure 2, the authors ascribe cause to a number of events without any supporting evidence. For instance, the authors state that the core of one of the particles dissolves until the solution is supersaturated. However, the authors don't know whether the halting of particle dissolution is caused by the solution becoming saturated. Similarly, they state that crack initiation leads to enhanced ion diffusion. It is not know that crack initiation is causes enhanced diffusion. I don't dispute that these dynamics are occurring, but the authors should adjust their language so they don't make such strong claims about processes they don't have supporting evidence for. Instead I think a better approach is to discuss these processes

together with supporting evidence (other characterization techniques, prior literature, etc). In situ TEM observations alone are not sufficient evidence to make claims about the solution chemistry because it cannot be directly seen.

Authors' Reply

Thanks for the comments. We have made clarification and modifications accordingly based on the comments.

For the statement of "... the metal core of P2 dissolves in the solution until the solution is saturated", we would like to mention that "the saturated solution" is evidenced by the newly precipitated CdCl₂ layer as highlighted by a yellow circle (Fig 2a: 15.7 s).

Even though we can't directly characterize the solution chemistry, comparing the area changes of core 1 and core 2 (Fig. 2c) indicates that the dissolution of core 2 promoted the growth of core 1. The dissolution of substances can increase the local ionic concentration in the solution, enhance the concentration difference, and then improve the diffusion of ions in the solution.

We have adjusted the language accordingly in the revised manuscripts to make our statement more clear and concise.

Comment 3: What causes the dissolution? Clearly the electron beam is causing this, but the authors should address the possible chemical mechanism inducing the crack formation and core dissolution. There are many times in the manuscript where it is implied that digestive ripening is driven by thermodynamics, but I expect that electron beam induced oxidation is driving the dissolution.

Authors' Reply

We agree that electron beam can enhance the dissolution of Cd. However, our experiment showed that Cd can still dissolve without the electron beam (Supplementary Fig. 9).

Since HCl was present in the solution, we propose that a possible reaction for the core to dissolve is:

Besides H^+ , electron beam irradiation of water may generate OH^\bullet or H_3O^+ , which can be the oxidants. Thus, the reaction becomes:

Or

In addition, the formation of cracks may occur spontaneously (Figure 5 in updated manuscript).

The possible mechanism is:

We agree the oxidation of Cd and the reduction of Cd^{2+} are driven by the reaction with H^+ and other radical species (induced by electron beam), but the mass transfer trend is driven by chemical potential that related to the oxidation of Cd. The oxidized metal cadmium ions cause the enhanced concentration of Cd ions, the change chemical potential drive mass transfer. We have rewritten the discussion part to describe our observation more accurately.

Comment 4: In the beginning of the discussion the authors state they estimate the concentration of Cd^{2+} in the solution. How is this done? The statement is kind of non-connected to this section of the discussion, so its unclear why this is mentioned. After this, they state that the low ion concentration in solution promotes Cd dissolution. This simply is not true; Cd is not soluble in the solvent and is thus not subject to thermodynamic driven Cd dissolution. As mentioned previously, I expect that the electron beam is promoting the Cd dissolution through an oxidation process.

Authors' Reply:

We assumed that the solution was in an enclosed space, and we estimated the concentration of Cd ions according to the change of particle morphology (See method part for details). We agree that Cd is almost insoluble in pure water solution, but the electron beam irradiation and HCl can promote the Cd dissolution. Cd metal can be oxidized into Cd^{2+} ions in the solution.

As shown in Supplementary Fig. 9, even without beam irradiation, the Cd particle can be etched away. The oxidation of Cd radical species and the reduction of Cd ions in the solution are reciprocal processes. The redox reactions are regulated by the chemical potential of ions, which is related to ions concentration. We have made modifications in the discussion part in the revised manuscript.

Comment 5: In general, the authors make several claims in the discussion about the relative concentration of Cd²⁺ in solution. But as mentioned above they don't know the Cd²⁺ concentration and can only infer it based on the dissolution kinetics of the Cd metal. Further, it's more complex than simply looking at Cd metal dissolution kinetics, because one must consider changes in the CdCl₂ layer as well. I think the discussion should be reworded in several places to remove mentions of the relative Cd²⁺ concentration because it cannot be discerned from their microscope images.

Authors' Reply:

Thanks for the comment. In the revised manuscript, we have made major changes in the discussion part and have removed expression of relative Cd²⁺ concentration.

Summary of the changes:

(Changes in the revised manuscript are highlighted in blue color)

1. In revised manuscript, page 1, line 3, the author “David T. Limmer” is added because he have contributions to Monte Carlo simulation and the revision of manuscript.
2. In revised manuscript, page 1, line 12, “Department of Chemistry, University of California, Berkeley, CA 94720, USA” is added. All affiliations are reordered.
3. In revised manuscript, page 1, line 13, “Chemical Science Division, Lawrence Berkeley National Laboratory, Berkeley, CA 94720, USA” is added. All affiliations are reordered.
4. In revised manuscript, page 1, line 14, “Kavli Energy Nanoscience Institute, Berkeley, CA 94720, USA” is added. All affiliations are reordered.
5. In revised manuscript, page 1, line 18, “, and thus they remain elusive” is deleted.
6. In revised manuscript, page 1, line 24, “At the end,” is changed to “Subsequent”.

7. In revised manuscript, page 2, line 29, “CdCl₂” is added before shell.
8. In revised manuscript, page 2, line 30, “Cd-CdCl₂” is added before CSN.
9. In revised manuscript, page 2, line 34, “R” is changed to “r”.
10. In revised manuscript, page 2, line 35, “free” is added before “energy”.
11. In revised manuscript, page 2, line 40, “many ripening processes are described as thermodynamic driven through minimization of the system energy” is changed to “many ripening processes are only described thermodynamically”.
12. In revised manuscript, page 2, line 40, “detailed” is added before “atomic pathways”.
13. In revised manuscript, page 3, line 73, “Fig. 1d, f, g” is changed to “Figs. 1d, f, g”.
14. In revised manuscript, page 4, line 74, “Supplementary Fig. 2, 3” is changed to “Supplementary Figs. 2, 3”.
15. In revised manuscript, page 4, line 84, “Fig. 2a” is changed to “Figs. 2a&b”.
16. In revised manuscript, page 4, line 87, “Fig. 2b” is changed to “Fig. 2c”.
17. In revised manuscript, page 4, line 89, “Fig. 2c” is changed to “Fig. 2d”.
18. In revised manuscript, page 4, line 90-91, “In stage I (0-16 s), only the core of P2 (core 2) dissolves in the solution until the solution is saturated.” is modified to “In stage I (0-16 s), only the metal core of P2 (core 2) is oxidized to Cd²⁺ and dissolves in the solution.”.
19. In revised manuscript, page 4, line 96, “Fig. 2a” is modified to “Fig. 2b”.
20. In revised manuscript, page 4, line 93, “(Fig. 2a: 15.7 s)” is modified to “(Fig. 2b: 15.7 s), which reflects the solution is CdCl₂ saturated.”.
21. In revised manuscript, page 4-5, line 97-98, “crack defects are formed in shell 1 while the shell expands” is modified to “crack defects form in shell 1 while the shell expands contributing to the growth of core 1”.
22. In revised manuscript, page 5, line 99, “Fig. 2a” is modified to “Fig. 2b”.
23. In revised manuscript, page 5, line 99, “from their anticorrelation” is added.
24. In revised manuscript, page 5, line 102, “Fig. 2c” is modified to “Fig. 2d”.
25. In revised manuscript, page 5, line 103, “Fig. 2b” is modified to “Fig. 2c”.
26. In revised manuscript, page 5, line 104, “Fig. 2a” is modified to “Fig. 2a&b”.
27. In revised manuscript, page 5, line 107, “nontraditional” is modified to “unique”.
28. In revised manuscript, page 5, line 108, “Fig. 2a” is modified to “Fig. 2b”.
29. In revised manuscript, page 5, line 110, “So, starting” is modified to “Starting”.
30. In revised manuscript, page 5, line 110, “the” is modified to “a”.
31. In revised manuscript, page 6, line 122, “Fig. 3a, b” is modified to “Figs. 3a, b”.
32. In revised manuscript, page 6, line 128, “Fig. 3b, d” is modified to “Figs. 3b, d”.
- 33.
34. In revised manuscript, page 6, line 130-131, “change the interface heterogeneity and” is

- deleted.
35. In revised manuscript, page 6, line 131, “the dissolution kinetics” is modified to “chemical potential”.
 36. In revised manuscript, page 6, line 135-136, “Similarly, the growth is mediated by the crack defects.” is modified to “As with dissolution of P2, growth of P1 is mediated by the crack defects.”.
 37. In revised manuscript, page 7, line 161, “driven by chemical potential difference,” is deleted.
 38. In revised manuscript, page 7, line 162, “free” is added before “energy”.
 39. In revised manuscript, page 7, line 164, “reduce” is modified to “reducing”.
 40. In revised manuscript, page 8, line 167-169, “Based on our in-situ observations, the defect-mediated ripening of Cd-CdCl₂ core-shell nanoparticles is illustrated in Fig. 5a. We estimated the Cd ion concentration variations in the solution during the ripening based on the shape change of nanoparticles, and the results are shown in Supplementary Fig. 9.” is deleted.
 41. In revised manuscript, page 8, line 170, “In the initial stage, the dissolution of Cd nanoparticles is favorable with low ion concentration.” is modified to “In the initial stage, the etching of Cd nanoparticles is driven by oxidation reaction.”.
 42. In revised manuscript, page 8, line 171, “dissolve, and the dissolution” is modified to “shrink, and the oxidative etching”.
 43. In revised manuscript, page 8, line 172-173, “heterogeneity difference at the solid-liquid interface results” is modified to “The preferential dissolution of Cd is kinetically facilitated by the broken shell.”.
 44. In revised manuscript, page 8, line 173, “the etching of” is added before “Cd”.
 45. In revised manuscript, “For example, there are... are shown in Fig. 5b.” in line 187-200 of the old version is deleted.
 46. In revised manuscript, “At the early stage of ripening... the ripening trend of nanoparticles is determined by the chemical potential difference.” in line 205-211 of the old version is deleted.
 47. In revised manuscript, page 8-9, line 189-218, “In order to understand the generality of the crack-mediated ripening mechanism ... and likely to happen in any core-shell construct.” is added.
 48. In revised manuscript, page 10, line 221, “Supplementary Fig. 11” is modified to “Supplementary Fig. 10”.
 49. In revised manuscript, page 10, line 225, “Supplementary Fig. 12” is modified to “Supplementary Fig. 11”.
 50. In revised manuscript, page 13, line 304-335, “Metropolis Monte Carlo simulation ... A

time-step is considered N attempted Monte Carlo moves.” is added.

51. In revised manuscript, page 15, line 352, “Supplementary Fig. 9” is modified to “Supplementary Fig. 15”.
52. In revised manuscript, page 15, line 383, “Supplementary Fig. 11 and 12” is modified to “Supplementary Fig. 10 and 11”.
53. In page 21, line 486, “37. Wu, F.Y. The potts model. *Rev. Mod. Phys.* 54, 235 (1982).” is added.
54. In page 21, line 487, “38. Chui, S.T. & Weeks, J.D., Dynamics of the roughening transition. *Phys. Rev. Lett.* 40, 733 (1978).” is added.
55. In page 21, line 488-489, “39. Frenkel, D., Smit, B., *Understanding molecular simulation: from algorithms to applications*. Vol. 1. Elsevier (2001).” is added.
56. In Author Contribution Part, page 22, line 503, “D. T. L. did the metropolis Monte Carlo simulation.” is added.
57. In Author Contribution Part, page 22, line 505, “D. T. L.” is added.
58. In Fig. 2, a schematic illustration figure added as Fig. 2a. In the legend, “Schematic illustration of the evolution pathway of Cd-CdCl₂ core-shell nanostructures during the defects-mediated ripening.” is added.
59. Fig. 5 and corresponding legend are added.
60. The reference is updated and reordered.

REVIEWER COMMENTS

Reviewer #1 (Remarks to the Author):

The authors have provided a rather detailed response to the comments of all three referees and clearly made an effort to improve their manuscript, including the addition of an entirely new model. As stated before, I consider the microscopy work excellent but I do not gain much insight from the modeling. One of the key criticisms has been the calculation of adsorption energies that are thermodynamic not really relevant, which has not been suitably addressed. I think, however, that the overall quality of this work is very good and should be published.

Reviewer #2 (Remarks to the Author):

I am satisfied with the response. Please make sure to address the issues brought up by reviewers either in the main text or SI. Some of the questions are only answered in the response but not addressed in the main text or SI.

Reviewer #3 (Remarks to the Author):

The authors have mostly addressed my concerns with the manuscript. The Monte carlo simulations address my major concerns. I have one more comment based on their response to comment 2, about how they know the solution is supersaturated. They base their inference of supersaturation on the formation of new CdCl₂ layers. However, this is a heterogeneous nucleation process, which can occur at undersaturated conditions. So in summary, using observations of nucleation to assess saturation is a qualitative metric and doesn't give any quantitative information about what the saturation level is. Saturation level necessary for nucleation varies whether it is homogeneous or heterogeneous nucleation, as well as what the heterogeneous surface is made of. After the authors address this minor comment, I recommend publication.

Response to the reviewers' comments

Reviewer #1 (Remarks to the Author):

The authors have provided a rather detailed response to the comments of all three referees and clearly made an effort to improve their manuscript, including the addition of an entirely new model. As stated before, I consider the microscopy work excellent but I do not gain much insight from the modeling. One of the key criticisms has been the calculation of adsorption energies that are thermodynamic not really relevant, which has not been suitably addressed. I think, however, that the overall quality of this work is very good and should be published.

Authors' Reply:

Thank you for the comments. In our new model, the influence of the solution and chemical potentials are considered. The simulation results show the defect-mediated ripening of core-shell particles is a kinetic process, which is consistent with our experimental observation. The new model can sufficiently support our claims. So, we have deleted the inadequate description on the adsorption energies calculation to avoid misunderstanding in this revised manuscript.

Reviewer #2 (Remarks to the Author):

I am satisfied with the response. Please make sure to address the issues brought up by reviewers either in the main text or SI. Some of the questions are only answered in the response but not addressed in the main text or SI.

Authors' Reply:

Thanks for the suggestion. We have updated the manuscript by adding additional materials from our responses to the review comments.

Reviewer #3 (Remarks to the Author):

The authors have mostly addressed my concerns with the manuscript. The Monte carlo simulations address my major concerns. I have one more comment based on their response to comment 2, about how they know the solution is supersaturated. They base their inference of

supersaturation on the formation of new CdCl₂ layers. However, this is a heterogeneous nucleation process, which can occur at under saturated conditions. So in summary, using observations of nucleation to assess saturation is a qualitative metric and doesn't give any quantitative information about what the saturation level is. Saturation level necessary for nucleation varies whether it is homogeneous or heterogeneous nucleation, as well as what the heterogeneous surface is made of. After the authors address this minor comment, I recommend publication.

Authors' Reply

Thanks for the valuable comments. We agree with these evaluation and clarification on saturation. We have made changes accordingly, in page 4 of the main text, we have changed the statement “supersaturated” to “saturation”. Now, it reads: “which reflects the solution is CdCl₂ saturated.”

Summary of the changes:

(Changes in the revised manuscript are highlighted in blue color)

1. In revised manuscript, page 1, line 1, the format of title change from bold to regular.
2. In revised manuscript, page 1, line 18, the format of “**Abstract**” change from bold to regular.
3. In revised manuscript, page 1, line 20, “**many**” is deleted.
4. In revised manuscript, page 1, line 25, “be” is modified to “**were**”.
5. In revised manuscript, page 2, line 30, the format of “**Introduction**” change from bold to regular.
6. In revised manuscript, page 2-3, line 45-46, “**Here, with cadmium-cadmium chloride (Cd-CdCl₂) CSN as a model system, we study the ripening of colloidal nanocrystals with a focus on the role of defects.**” is moved from the second paragraph to the third paragraph of introduction.
7. In revised manuscript, page 2, line 46, “The Cd-CdCl₂ core-shell” is modified to “**Core-shell**”.
8. In revised manuscript, page 3, line 62, “**Results**” is modified to “**Results and Discussion**”.
9. In revised manuscript, page 4, line 75, “**Supplementary Table 1**” is added after “Supplementary Fig. 2, 3”.
10. In revised manuscript, page 4, line 83, the format of “**The dynamic of defect-mediated ripening**” change from bold to regular.
11. In revised manuscript, page 4, line 97-98, “**which reflects the solution is CdCl₂ supersaturated.**” is deleted.
12. In revised manuscript, page 5, line 117, the format of “**The impact of crack defects on ripening process**” change from bold to regular.
13. In revised manuscript, page 7, line 162, “oversaturated” is modified to “**saturated**”.
14. In revised manuscript, page 8, line 167-173, “**Given the crystallographic...but these defects are quickly repaired.**” is added.
15. In revised manuscript, page 8, line 176, the format of “**Discussion**” is deleted.
16. In revised manuscript, page 9, line 188, the format of equation is modified.
17. In revised manuscript, page 10, line 234, the format of “**Conclusion**” is deleted.
18. In revised manuscript, page 11, line 244, the format of “**Methods**” change from bold to regular.
19. In revised manuscript, page 11, line 252, the format of “**CdS nanorods preparation**” change from bold to regular.
20. In revised manuscript, page 12, line 257, “6000 r.p.m.” is modified to “**5647 g**”.
21. In revised manuscript, page 12, line 259, the format of “**In situ liquid cell experimental setup**” change from bold to regular.
22. In revised manuscript, page 12, line 268, the format of “**Formation of Cd-CdCl₂ core-shell nanostructures**” change from bold to regular.

23. In revised manuscript, page 13, line 280-284, the format of equation is modified.
24. In revised manuscript, page 13, line 285, the format of “**The determination of a CdCl₂ shell covering on the (0001) surface of the core.**” change from bold to regular.
25. In revised manuscript, page 13, line 296, the format of equation is modified.
26. In revised manuscript, page 14, line 300-315, “**The density ...molecule in a vacuum**” is deleted.
27. In revised manuscript, page 16, line 349, the format of “**Estimation of ion concentration in the solution**” change from bold to regular.
28. In revised manuscript, page 16, line 360, “were 13000 nm³ and” is modified to “was”.
29. In revised manuscript, page 17, line 365, “Table 3” is modified to “**Supplementary Table 2**”.
30. In revised manuscript, page 17, line 366-367, “Defect-mediated ripening is a new phenomenon” is modified to “**Defect-mediated ripening is different from any reported ripening phenomenon**”.
31. In revised manuscript, page 18, line 392, the format of “**Defect-driven growth can achieve unique growth**” change from bold to regular.
32. In revised manuscript, page 18, line 408, the format of “**Exclude the influence of surrounding particles on the ripening of the nanoplate pair**” change from bold to regular.
33. In revised manuscript, page 19, line 424-433, “**The influence of electron beam ...crack generation in this experiment**” is added.
34. In revised manuscript, page 20, line 435-438, “**Data available...on request**” is added.
35. In revised manuscript, page 20, line 440-442, “**Code available...upon request**” is added.
36. In revised manuscript, page 23, references 34-36 are deleted.
37. In revised manuscript, page 24, line 537, the format of “**Competing interests**” change from bold to regular.
38. In revised manuscript, page 24, line 539, “**Competing interests**” change from bold to regular.
39. In revised manuscript, page 24, line 539-541, “**Data Availability... reasonable request.**” is deleted.
40. In the caption of Fig. 1, page 24, “STEM-EDS elemental mapping of Cd-CdCl₂ core-shell structures” is modified to “**High-angle annular dark-field imaging (HAADF) and energy-dispersive X-ray spectroscopy (EDS) mapping of Cd-CdCl₂ core-shell structures.**”
41. In the caption of Fig. 2, page 25, “**I , II, III, and IV refer to the four intervals of the dynamic process**” is added.
42. In the caption of Fig. 3, page 25, “**The yellow arrows point out the gap between core and shell**” is added.
43. In the caption of Fig. 4, page 25, “**Symbol ‘T’ represents edge dislocation**” is added.
44. In the caption of Fig. 4, page 25, “**Arabic numerals mark the order of the CdCl₂ shells.**” is added.

In the caption of Fig. 5, page 26, “The shadings around the solid lines indicate the standard deviation.” is added.

45. The supplementary figures are reordered.

46. The reference is updated and reordered.